# Investigating and Simplifying Masking-based Saliency Map Methods for Model Interpretability

## Abstract

Saliency maps that identify the most informative regions of an image for a classifier are valuable for model interpretability. A common approach to creating saliency maps involves generating input masks that mask out portions of an image to maximally deteriorate classification performance, or mask in an image to preserve classification performance. Many variants of this approach have been proposed in the literature, such as counterfactual generation and optimizing over a Gumbel-Softmax distribution. Using a general formulation of masking-based saliency methods, we conduct an extensive evaluation study of a number of recently proposed variants to understand which elements of these methods meaningfully improve performance. Surprisingly, we find that a well-tuned, relatively simple formulation of a masking-based saliency model outperforms many more complex approaches. We find that the most important ingredients for high quality saliency map generation are (1) using both masked-in and masked-out objectives and (2) training the classifier alongside the masking model. Strikingly, we show that a masking model can be trained with as few as 10 examples per class and still generate saliency maps with only a 0.7-point increase in localization error.

## 1 Introduction

The success of CNNs (Krizhevsky et al., 2012; Szegedy et al., 2015; He et al., 2016; Tan & Le, 2019) has prompted interest in improving understanding of how these models make their predictions. Particularly in applications such as medical diagnosis, having models explain their predictions can improve trust in them. The main line of work concerning *model interpretability* has focused on the creation of saliency maps–overlays to an input image that highlight regions most salient to the model in making its predictions. Among these, the most prominent are gradient-based methods (Simonyan et al., 2013; Sundararajan et al., 2017; Selvaraju et al., 2018) and masking-based methods (Fong & Vedaldi, 2017; Dabkowski & Gal, 2017; Fong & Vedaldi, 2018; Petsiuk et al., 2018; Chang et al., 2019; Zintgraf et al., 2017). In recent years, we have witnessed an explosion of research based on these two directions. With a variety of approaches being proposed, framed and evaluated in different ways, it has become difficult to assess and fairly evaluate their additive contributions.

In this work, we investigate the class of masking-based saliency methods, where we train a masking model to generate saliency maps based on an explicit optimization objective. Using a general formulation, we iteratively evaluate the extent to which recently proposed ideas in the literature improve performance. In addition to evaluating our models against the commonly used Weakly Supervised Object Localization (WSOL) metrics, the Saliency Metric (SM), and the more recently introduced Pixel Average Precision (PxAP; Choe et al., 2020), we also test our final models against a suite of "sanity checks" for saliency methods (Adebayo et al., 2018; Hooker et al., 2018).

Concretely, we make four major contributions. (1) We find that incorporating both masked-in classification maximization and masked-out entropy maximization objectives leads to the best saliency maps, and continually training the classifier improves the quality of generated maps. (2) We find that the masking model requires only the top layers of the classifier to effectively generate saliency maps. (3) Our final model outperforms other masking-based methods on WSOL and PxAP metrics. (4) We find that a small number of examples—as few as ten per class—is sufficient to train a masker to within the ballpark of our best performing model.

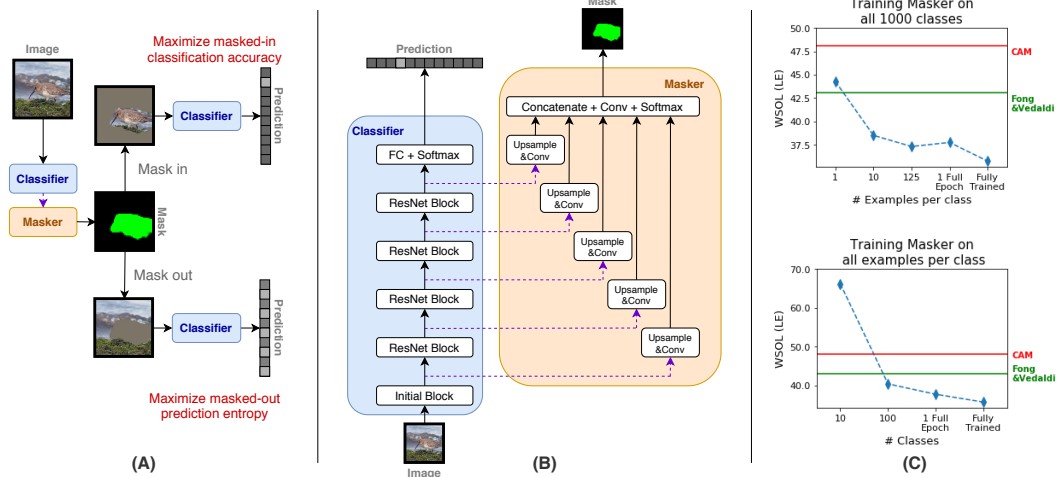

Figure 1: (A) Overview of the training setup for our final model. The masker is trained to maximize masked-in classification accuracy and masked-out prediction entropy. (B) Masker architecture. The masker takes as input the hidden activations of different layers of the ResNet-50 and produces a mask of the same resolution as the input image. (C) Few-shot training of masker. Performance drops only slightly when trained on much fewer examples compared to the full training procedure.

## 2 RELATED WORK

Interpretability of machine learning models has been an ongoing topic of research (Ribeiro et al., 2016; Doshi-Velez & Kim, 2017; Samek et al., 2017; Lundberg et al., 2018). In this work, we focus on interpretability methods that involve generating saliency maps for image classification models. An overwhelming majority of the methods for generating saliency maps for image classifiers can be assigned to two broad families: gradient-based methods and masking-based methods.

**Gradient-based methods**, such as using backpropagated gradients (Simonyan et al., 2013), Guided Backprop (Springenberg et al., 2015), Integrated Gradients (Sundararajan et al., 2017), GradCam (Selvaraju et al., 2018), SmoothGrad (Smilkov et al., 2017) and many more, directly use the backpropagated gradients through the classifier to the input to generate saliency maps.

**Masking-based methods** modify input images to alter the classifier behavior and use the regions of modifications as the saliency map. Within this class of methods, one line of work focuses on optimizing over the masks directly: Fong & Vedaldi (2017) optimize over a perturbation mask for an image, Petsiuk et al. (2018) aggregates over randomly sampled masks, Fong & Vedaldi (2018) performs an extensive search for masks of a given size, while Chang et al. (2019) includes a counterfactual mask-infilling model to make the masking objective more challenging. The other line of work trains a separate masking model to produce saliency maps: Dabkowski & Gal (2017) trains a model that optimizes similar objectives to Fong & Vedaldi (2017), Zolna et al. (2020) use a continually trained pool of classifiers and an adversarial masker to generate model-agnostic saliency maps, while Fan et al. (2017) identifies super-pixels from the image and then trains the masker similarly in an adversarial manner.

**Salient Object Detection** (Borji et al., 2014; Wang et al., 2019) is a related line of work that concerns identifying salient objects within an image as an end in itself, and not for the purpose of model interpretability. While it is not uncommon for these methods to incorporate a pretrained image classification model to extract learned visual features, they often also incorporate techniques for improving the quality of saliency maps that are orthogonal to model interpretability. Salient object detection methods that are trained on only image-level labels bear the closest similarity to saliency map generation methods for model interpretability. Hsu et al. (2017) and follow-up Hsu et al. (2019) train a masking model to confuse a binary image-classification model that predicts whether an image contains an object or is a 'background' image. Wang et al. (2017) apply a smooth pooling operation

and a Foreground Inference Network (a masking model) while training an image classifier to generate saliency maps as a secondary output.

**Evaluation of saliency maps** The wave of saliency map research has also ignited research on evaluation methods for these saliency maps as model explanations. Adebayo et al. (2018) and Hooker et al. (2018) propose sanity checks and benchmarks for the saliency maps. Choe et al. (2020) propose Pixel Average Precision (PxAP), a pixel-wise metric for scoring saliency maps that accounts for mask binarization thresholds, while Yang & Kim (2019) create a set of metrics as well as artificial datasets interleaving foreground and background objects for evaluating the saliency maps. These works have shown that a number of gradient-based methods fail the sanity checks or perform no better than simple edge detectors. Hence, we choose to focus on masking-based methods in this paper.

## 3 Masking-based Saliency Map Methods

We start by building a general formulation of masking-based saliency map methods. We take as given a trained image classifier $F : x \rightarrow y$, that maps from image inputs $x \in \mathbb{R}^{H \times W \times C}$ to class predictions $\hat{y} \in [0, 1]^K$, evaluated against ground-truth $y \in \{1 \cdots K\}$. Our goal is to generate a mask $m \in [0, 1]^{H \times W}$ for each image $x$ such that the masked-in image $x \odot m$ or the masked-out image $x \odot (1 - m)$ maximizes some objective based on output of a classifier given the modified image. For instance, we could attempt to mask out parts of the image to maximally deteriorate the classifier's performance. This mask $m$ then serves as a saliency map for the image $x$. Concretely, the per-image objective can be expressed as:

$$\underset{m}{\arg\min} \; \lambda_{\text{out}} L_{\text{out}}\big(F(x \odot (1 - m); \theta_F), y\big) + \lambda_{\text{in}} L_{\text{in}}\big(F(x \odot m; \theta_F), y\big) + R(m),$$

where $L_{\text{out}}, L_{\text{in}}$ are the masked-out and masked-in objectives over the classifier output, $\lambda_{\text{out}}, \lambda_{\text{in}}$ are hyperparameters controlling weighting of these two objectives, $\theta_F$ the classifier parameters, and $R(m)$ a regularization term over the mask. The masked-in and masked-out losses, $L_{\text{out}}$ and $L_{\text{in}}$, correspond to finding the smallest destroying region and smallest sufficient region as described in Dabkowski & Gal (2017). Candidates for $L_{\text{out}}$ include negative classification cross-entropy and prediction entropy. For $L_{\text{in}}$, the obvious candidate is the classification cross-entropy of the masked-in image. We set $\lambda_{\text{in}} = 0$ or $\lambda_{\text{out}} = 0$ if we only have either a masked-in or masked-out objective.

The above formulation subsumes a number of masking-based methods, such as Fong & Vedaldi (2017); Dabkowski & Gal (2017); Zolna et al. (2020). We follow Dabkowski & Gal, amortize the optimization by training a neural network masker $M : x \rightarrow m$, and solve for:

$$\underset{\theta_M}{\arg\min} \; \lambda_{\text{out}} L_{\text{out}}\big(F(x \odot (1 - M(x; \theta_M)); \theta_F), y\big) + \lambda_{\text{in}} L_{\text{in}}\big(F(x \odot M(x; \theta_M); \theta_F), y\big) + R(M(x; \theta_M)),$$

where $M$ is the masking model and $\theta_M$ its parameters. In our formulation, we do not provide the masker with the ground-truth label, which differs from certain other masking-based saliency works (Dabkowski & Gal, 2017; Chang et al., 2019; Fong & Vedaldi, 2018). In practice, we often desire model explanations without the availability of ground-truth information, so we focus our investigation on methods that require only an image as input.

### 3.1 Masker Architecture

We use a similar architecture to Dabkowski & Gal and Zolna et al.. The masker takes as input activations across different layers of the classifier, meaning it has access to the internal representation of the classifier for each image. Each layer of activations is fed through a convolutional layer and upsampled (with nearest neighbor interpolation) so they all share the same spatial resolution. All transformed layers are then concatenated and fed through another convolutional layer, upsampled, and put through a sigmoid operation to obtain a mask of the same resolution as the input image. In all our experiments, we use a ResNet-50 (He et al., 2016) as our classifier, and the masker has access to the outputs of the five major ResNet blocks. Figure 1B shows the architecture of our models.

Following prior work (Fong & Vedaldi, 2017), we apply regularization on the generated masks to avoid trivial solutions such as masking the entire image. We apply L1 regularization to limit the size of masks and Total Variation (TV) to encourage smoothness. Details can be found in Appendix A.1.

| Model | | | | OM ↓ | LE ↓ | SM ↓ | PxAP ↑ |
|---|---|---|---|---|---|---|---|
| *Train-Validation Set* | | | | | | | |
| a) FIX | + MaxEnt (O) | | | 49.8± 0.31 | 39.4± 0.35 | 0.22± 0.015 | 48.8± 1.00 |
| b) FIX | + MinClass (O) | | | 62.5± 4.29 | 54.8± 5.15 | 0.25± 0.170 | 45.6± 6.67 |
| c) FIX | + MaxClass (I) | | | 50.3± 0.04 | 39.9± 0.04 | 0.14± 0.001 | 55.4± 0.10 |
| d) CA | + MaxEnt (O) | | | 46.7± 0.13 | 34.9± 0.16 | 0.21± 0.004 | 51.3± 0.16 |
| e) CA | + MinClass (O) | | | 45.9± 0.06 | 34.1± 0.05 | 0.17± 0.004 | 54.8± 0.18 |
| f) CA | + MaxClass (I) | | | 55.4± 4.76 | 45.7± 5.30 | 0.30± 0.044 | 43.1± 3.72 |
| g) FIX | + MaxClass (I) + MaxEnt (O) | | | 46.7± 0.02 | 35.7± 0.02 | 0.07± 0.002 | 57.0± 0.09 |
| h) FIX | + MaxClass (I) + MinClass (O) | | | 50.6± 1.70 | 40.2± 2.03 | 0.14± 0.064 | 51.3± 3.98 |
| i) CA | + MaxClass (I) + MaxEnt (O) | | | 45.0± 0.06 | 33.4± 0.06 | 0.12± 0.004 | 60.2± 0.08 |
| j) CA | + MaxClass (I) + MinClass (O) | | | 45.6± 0.09 | 34.1± 0.10 | 0.17± 0.004 | 60.6± 0.14 |
| k) FIX | + MaxClass (I) + MaxEnt (O) | + Layer[1] | | 57.3± 0.07 | 48.3± 0.09 | 0.40± 0.005 | 34.7± 0.20 |
| l) FIX | + MaxClass (I) + MaxEnt (O) | + Layer[3] | | 53.6± 0.03 | 43.8± 0.04 | 0.31± 0.001 | 44.0± 0.10 |
| m) FIX | + MaxClass (I) + MaxEnt (O) | + Layer[5] | | 48.4± 0.03 | 37.8± 0.05 | 0.04± 0.001 | 55.5± 0.15 |
| n) FIX | + MaxClass (I) + MaxEnt (O) | + Layer[4–5] | | 47.0± 0.05 | 36.0± 0.07 | 0.04± 0.001 | 58.2± 0.13 |
| o) CA | + MaxClass (I) + MaxEnt (O) | + Layer[1] | | 75.3± 0.73 | 69.3± 0.88 | 0.49± 0.008 | 27.2± 0.00 |
| p) CA | + MaxClass (I) + MaxEnt (O) | + Layer[3] | | 55.4± 0.39 | 45.7± 0.49 | 0.27± 0.012 | 44.3± 0.17 |
| q) CA | + MaxClass (I) + MaxEnt (O) | + Layer[5] | | 46.6± 0.08 | 35.4± 0.09 | 0.11± 0.003 | 56.6± 0.22 |
| r) CA | + MaxClass (I) + MaxEnt (O) | + Layer[4–5] | | 45.1± 0.04 | 33.3± 0.05 | 0.11± 0.002 | 61.1± 0.13 |
| s) FIX | + MaxClass (I) + MaxEnt (O) | + Layer[4–5] | + Inf[Blur] | 47.1± 0.04 | 36.2± 0.04 | 0.08± 0.002 | 58.7± 0.07 |
| t) FIX | + MaxClass (I) + MaxEnt (O) | + Layer[4–5] | + Inf[CAG] | 49.7± 0.05 | 39.2± 0.06 | 0.11± 0.002 | 52.9± 0.15 |
| u) FIX | + MaxClass (I) + MaxEnt (O) | + Layer[4–5] | + Inf[DFN] | 49.9± 0.03 | 39.5± 0.04 | -0.03± 0.001 | 55.2± 0.08 |
| v) CA | + MaxClass (I) + MaxEnt (O) | + Layer[4–5] | + Inf[Blur] | 46.0± 0.12 | 34.5± 0.13 | 0.07± 0.004 | 59.5± 0.19 |
| w) CA | + MaxClass (I) + MaxEnt (O) | + Layer[4–5] | + Inf[CAG] | 45.4± 0.05 | 33.6± 0.05 | 0.12± 0.002 | 56.7± 0.08 |
| x) CA | + MaxClass (I) + MaxEnt (O) | + Layer[4–5] | + Inf[DFN] | 49.0± 0.12 | 38.3± 0.14 | 0.02± 0.001 | 61.1± 0.14 |
| y) FIX | + MaxClass (I) + MaxEnt (O) | + Layer[4–5] + Inf[CAG] | + GS | 52.8± 0.24 | 43.0± 0.30 | 0.19± 0.003 | 40.6± 0.20 |
| z) CA | + MaxClass (I) + MaxEnt (O) | + Layer[4–5] + Inf[CAG] | + GS | 48.9± 0.28 | 38.1± 0.30 | -0.01± 0.004 | 49.3± 0.16 |
| A) Mask In Everything | | | | 50.9 | 50.9 | 0.44 | 27.2 |
| B) Mask In Nothing | | | | 100.0 | 100.0 | 4.72 | 27.2 |
| C) Mask In Center 50% Area | | | | 68.1 | 68.1 | **-0.09** | 36.7 |
| *Validation Set* | | | | | | | |
| D) Fong & Vedaldi (2017) | | | | - | 43.1 | - | - |
| E) Dabkowski & Gal (2017) | | | | - | 36.9 | 0.32 | - |
| F) Zolna et al. (2020) | | | | 48.6 | 36.1 | - | - |
| G) Chang et al. (2019) | | | | - | 57.0 | **-0.02** | - |
| H) CAM | | | | - | 48.1 | - | 58.0 |
| I) Our Best FIX | | | | 51.5 | 39.7 | 0.59 | 54.5 |
| J) Our Best CA | | | | **48.4** | **35.8** | 0.52 | **59.4** |

Table 1: Evaluation of masking-based saliency map methods. Each block captures one set of experiments. FIX indicates a fixed classifier, CA (Classifier-Agnostic) indicates training against a pool of continually trained classifiers. MaxClass (I), MinClass (O) and MaxEnt (O) are masked-in classification-maximization, masked-out classification-minimization and masked-out entropy maximization objectives for the masker. Layer[·] indicates the layer or layers of classifier activations provided as input to the masker. Inf[·] indicates the infiller operation applied after masking–the default otherwise is no infilling. Columns show mean and standard errors over 5 runs for evaluation metrics Official Metric (OM) and Localization Error (LE) for weakly supervised localization, Saliency Metric (SM) and Pixel Average Precision (PxAP). Underlined results are the best results within that block, while **bold** are the best results for data set, excluding baselines.

## 3.2 CONTINUAL TRAINING OF THE CLASSIFIER

Because neural networks are susceptible to adversarial perturbations (Goodfellow et al., 2015), masking models can learn to perturb an input to maximize the above objectives for a given fixed classifier without producing intuitive saliency maps. While directly regularizing the masks is one potential remedy, Zolna et al. (2020) propose to train the masker against a diverse set of classifiers. In practice, they simulate this by continually training the classifier on masked images, retain a pool of past model checkpoints, and sample from the pool when training the masker.

We adopt their approach and distinguish between a masker trained against a fixed classifier (**FIX**) and against a pool of continually trained classifiers (**CA**, for Classifier-Agnostic). We highlight that saliency maps for FIX and CA address fundamentally different notions of saliency. Whereas a FIX approach seeks a saliency map that explains what regions are most salient to a given classifier, a CA approach tries to identify all possible salient regions for any hypothetical classifier (hence, classifier-agnostic). In other words, a CA approach may be inadequate for interpreting a specific classifier and is better suited for identifying salient regions for a class of image classification models.

# 4 EXPERIMENTAL SETUP

We perform our experiments on the official ImageNet training and validation set (Deng et al., 2009) and use bounding boxes from the ILSVRC'14 localization task. Because we perform a large number of experiments with hyperparameter search to evaluate different model components, we construct a separate held-out validation set of 50,000 examples (50 per class) from the training set with bounding box data that we use as validation for the majority of our experiments (which we refer to as our "Train-Validation" set) and use the remainder of the training set for training. For each model configuration, we train the models 5 times on different random seeds and report the mean and standard error of the results. We reserve the official validation set for the final evaluation.

## 4.1 EVALUATION METRICS

**Weakly-supervised object localization task metrics** (WSOL) is a common task for evaluating saliency maps. It involves generating bounding boxes for salient objects in images and scoring them against the ground-truth bounding boxes. To generate bounding boxes from our saliency maps, we binarize the saliency map based on the average mask pixel value and use the tightest bounding box around the largest connected component of our binarized saliency map. We follow the evaluation protocol in ILSVRC '14 computing the official metric (OM), localization error (LE) and pixel-wise F1 score between the predicted and ground-truth bounding boxes.

**Saliency metric** (SM) proposed by Dabkowski & Gal (2017) consists of generating a bounding box from the saliency map, upsampling the region of the image within the bounding box and then evaluating the classifier accuracy on the upsampled salient region. The metric is defined as $s(a, p) = \log(\max(a, 0.05)) - \log(p)$, where $a$ is the size of the bounding box, and $p$ is the probability the classifier assigns to the true class. This metric can be seen as measuring masked-in and upsampled classification accuracy with a penalty for the mask size. We use the same bounding boxes as described in WSOL for consistency.

**Pixel Average Precision** (PxAP) proposed by Choe et al. (2020) scores the pixel-wise masks against the ground-truth masks and computes the area under the precision-recall curve. This metric is computed over mask pixels rather than bounding boxes and removes the need to threshold and binarize the mask pixels. PxAP is computed over the OpenImages dataset (Benenson et al., 2019) rather than ImageNet because it requires pixel-level ground-truth masks.

# 5 EVALUATION OF SALIENCY MAP METHODS

To determine what factors and methods contribute to improved saliency maps, we perform a series of evaluation experiments in a cascading fashion. We isolate and vary one design choice at a time, and use the optimal configuration from one set of experiments in all subsequent experiments. Our baseline models consist of a masker trained with either a fixed classifier (FIX) or a pool of continually trained classifiers (CA). As WSOL is the most common task for evaluating saliency maps, we use LE as the metric for determining the 'best' model for model selection. We show our model scores across experiments in Table 3. Each horizon block represents a set of experiments varying one design choice. The top half of the table is evaluated on our Train-Validation split, while the bottom half is evaluated on the validation data from ILSVRC '14.

**Masking Objectives** (Rows `a-f`) We first consider varying the masker's training objective, using only one objective at a time. We use the three candidate objectives described in Section 3: maximizing masked-in accuracy, minimizing masked-out accuracy and maximizing masked-out entropy. For a masked-out objective, we set $\lambda_{\text{out}} = 1, \lambda_{\text{in}} = 0$, and the opposite for masked-in objectives. For each configuration, we perform a random hyperparameter search over the L1 mask regularization coefficients $\lambda_M$ and $\lambda_{TV}$ as well as the learning rate and report results from the best configuration from the Train-Validation set. More details on hyperparameter choices can be found in Table 2.

Consistent with Zolna et al. (2020), we find that training the classifier along with the masker improves the masker, with CA models generally outperforming FIX models, particularly for the WSOL metrics. However, the classification-maximization FIX model still performs comparably with its CA counterpart and in fact performs best overall when measured by SM given the similarity between the training objective and the second term of the SM metric. Among the CA models, entropy-maximization and classification-minimization perform the best, while the classification-maximization

objective performs worst. On the other hand, both mask-out objectives perform extremely poorly for a fixed classifier. We show how different masking objectives affect saliency map generation in Figure 2.

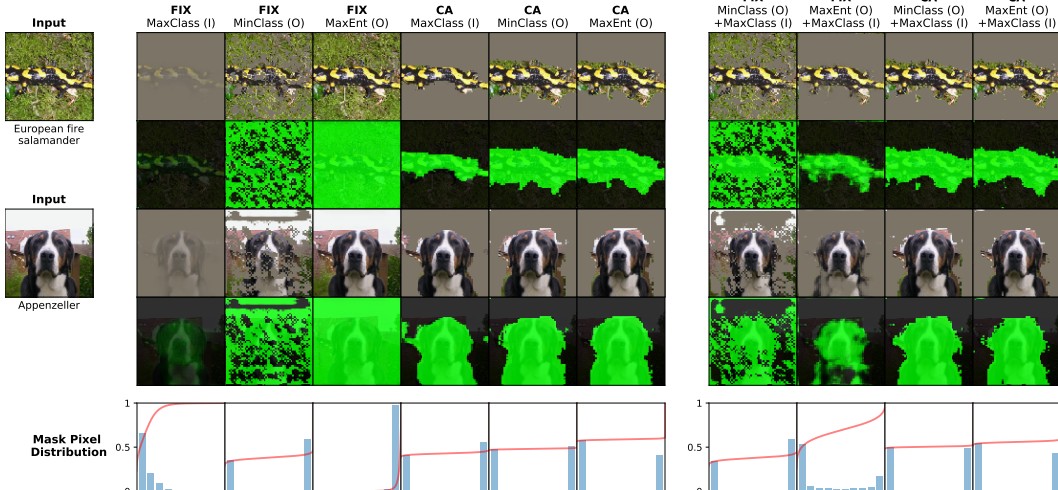

Figure 2: Varying the masking objective for a fixed classifier (FIX) and classifier-agnostic (CA) settings. **Top**: Examples of masked-in inputs and masks. The CA models generally produce more contiguous masks, and combining both a masked-in and masked-out objective works best based on quantitative evaluation. **Bottom**: Distribution of mask pixel values across Train-Validation set. We quantize the values into 10 buckets in the bar plot in blue and also show the empirical CDF in red. Most models produce highly bimodal distributions, with most pixel values close to 0 or 1.

**Combining Masking Objectives** (Rows g–j) Next, we combine both masked-in and masked-out objective during training, setting $\lambda_{out} = \lambda_{in} = 0.5$. Among the dual-objective models, entropy-maximization still outperforms classification-minimization as a masked-out objective. Combining both masked-in classification-maximization and masked-out entropy-maximization performs best for both FIX and CA models, consistent with Dabkowski & Gal (2017). From our hyperparameter search, we also find that separately tuning $\lambda_{M,in}$ and $\lambda_{M,out}$ is highly beneficial (see Table 2). We use the classification-maximization and entropy-maximization dual objectives for both FIX (Row g) and CA (Row i) models in subsequent experiments.

**Varying Observed Classifier Layers** (Rows k–r) We now vary which hidden layers of the classifier the masker has access to. As described in Section 3.1, the masking model has access to hidden activations from five different layers of a ResNet-50 classifier. To identify the contribution of information from each layer to the masking model, we train completely new masking models with access to only a subset of the classifier layers. We number the layers from 1 to 5, with 1 being the earliest layer with the highest resolution ($56 \times 56$) and 5 being the latest ($7 \times 7$). We show a relevant subset of the results from varying the observed classifier layers in Table 3. The full results can be found in the Table 4 and we show examples of the generated masks in Figure 3.

Masking models with access to activations of later layers starkly outperform those using activations from earlier layers. Whereas the Layer[3], Layer[4] and Layer[5] models are still effective, the Layer[1] and Layer[2] models tend to perform poorly. Similarly, we find that the best cascading combination of layers is layers 4 and 5 CA models, and 3–5 for FIX models (Rows n, r), slightly but consistently outperforming the above models with all layers available to the masker. This suggests that most of the information relevant for generating saliency maps is likely contained within the later layers. For simplicity, we use only classifier layers 4 and 5 for subsequent experiments.

**Counterfactual Infilling and Binary Masks** (Rows s–z) Chang et al. (2019) proposed generating saliency maps by learning a Bernoulli distribution per masking pixel and additionally incorporating a counterfactual infiller. Agarwal & Nguyen (2019) similarly uses an infiller when producing saliency maps. First, we consider applying counterfactual infilling to the masked images before feeding them to the classifier. The modified inputs are $\text{Infill}(X \odot (1 - m), (1 - m))$ and $\text{Infill}(X \odot m, m)$ for

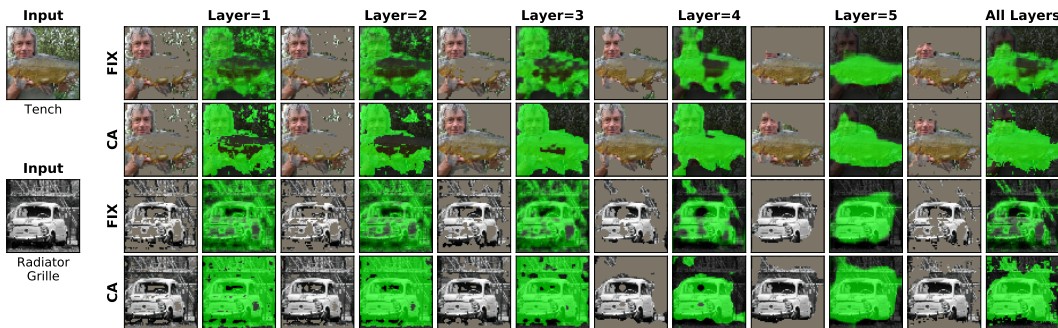

Figure 3: Examples of varying the layers of classifier activations that the masker takes as input, from Layers 1 (closest to input) to 5 (closest to output). Each column is from a separately trained model. Using activations from higher layers leads to better saliency maps, despite having lower resolution. *All Layers* has access to all 5 layers, but does not perform better than simply using layers 4 and 5.

masked-out and masked-in infilling respectively, where Infill is the infilling function that takes as input the masked input as well as the mask. We consider three infillers: the Contextual Attention GAN (Yu et al., 2018) as used in Chang et al.[1], DFNet (Hong et al., 2019), and a Gaussian blur infiller as used in Fong & Vedaldi (2018). Both neural infillers are pretrained and frozen.

For each infilling model, we also train a model variant that outputs a discrete binary mask by means of a Gumbel-Softmax layer (Jang et al., 2017; Maddison et al., 2017). We experiment with both soft and hard (Straight-Through) Gumbel estimators, and temperatures of $\{0.01, 0.05, 0.1, 0.5\}$.

We show a relevant subset of the results in the fourth and fifth blocks of Table 3, and examples in Figure 6 and Figure 7. We do not find major improvements from incorporating infilling or discrete masking based on WSOL metrics, although we do find improvements from using the DFN infiller for SM. Particularly for CA, because the classifier is continually trained to classify masked images, it is able to learn to both classify unnaturally masked images as well as to perform classification based on masked-out evidence. As a result, the benefits of incorporating the infiller may be diminished.

## 5.1 EVALUATION ON VALIDATION SET

Based on the above, we identify a simple recipe for a good saliency map generation model: (1) use both masked-in classification maximization and masked-out entropy maximization objectives, (2) use only the later layers of the classifier as input to the masker, and (3) continually train the classifier. To validate the effectiveness of this simple setup, we train a new pair of FIX and CA models based on this configuration on the full training set and evaluate on the actual ILSVRC '14 validation set. We compare the results to other models in the literature in the bottom block of Table 3. Consistent with above, the CA model outperforms the FIX model. It also outperforms other saliency map extraction methods on WSOL metrics and PxAP. We highlight that some models we compare to (Rows E, G) are provided with the ground-truth target class, whereas our models are not–this may explain the underperformance on certain metrics such as SM, which is partially based on classification accuracy.

## 5.2 SANITY CHECKS

Adebayo et al. (2018) and Hooker et al. (2018) propose "sanity checks" to verify whether saliency maps actually reflect what information classifiers use to make predictions and show that many proposed interpretability methods fail these simple tests. We apply these tests to our saliency map models to verify their efficacy. On the left of Figure 4, we show the RemOve-and-Retrain (ROaR) test proposed by Hooker et al., where we remove the top $t\%$ of pixels from training images based on our generated saliency maps and use them to train entirely new classifiers. If our saliency maps truly identify salient portions of the image, we should see large drops in classifier performance as $t$ increases. Both FIX and CA methods pass this test, with classifier accuracy falling precipitously as we mask out more pixels. On the right of Figure 4, we perform the the Model Parameter Randomization Test (MPRT) proposed by Adebayo et al.. We randomize parameters of successive layers of the

---

[1] The publicly released CA-GAN is only trained on rectangular masks, but Chang et al. nevertheless found positive results from applying it, so we follow their practice. DFNet is trained on irregularly shaped masks.

classifier, starting from upper-most logits layer to the lowest convolutional layers, and generate saliency maps using the partially randomized classifiers. We then compute the similarity of the saliency maps generated from using the partially randomized classifier, and those using the original classifier. Our saliency maps become less similar as more layers are randomized, passing the test. The results for the Data Randomization Test (DRT) can be found in Table 5.

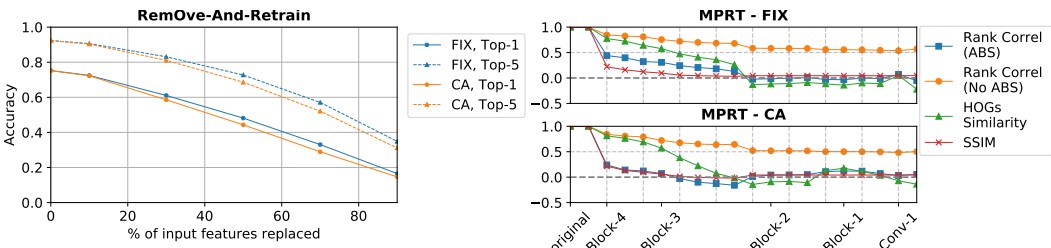

Figure 4: Our saliency maps pass recently proposed sanity checks. Left: RemOve-And-Retrain evaluation from Hooker et al. (2018). We train and evaluate new ResNet-50 models based on training images with the top $t\%$ of most salient pixels removed. Removing the most salient pixels hurts the new classifier's performance, and the CA model is more effective at removing salient information. Right: Model Parameter Randomization Test (MPRT) from Adebayo et al. (2018). We randomize parameters of successive layers of the classifier, and compute the similarity between the saliency maps generated using the original and modified classifier. Similarity falls as more layers are randomized.

## 6 FEW-SHOT EXPERIMENTS

Given the relative simplicity of our best-performing saliency map models and the fact that the masker uses only the top layers of activations from the classifier, we hypothesize that learning to generate saliency maps given strong classifier is a relatively simple process.

To test this hypothesis, we run a set of experiments severely limiting the number of training steps and unique examples that the masker is trained on. The ImageNet dataset consists of 1,000 object classes, with up to 1,250 examples per class. We run a set of experiments restricting both the number of unique classes seen as well as the number of examples seen per class while training the masker. We also limit the number of training steps be equivalent to one epoch through the full training set. Given the randomness associated with subsampling the examples, we randomly subsample classes and/or examples 5 times for each configuration and compute the median score over the 5 runs. We report results on the actual validation set for ILSVRC '14 (LE) and test set for OpenImages (PxAP). Examples of saliency maps for these models can be found in Figure 9.

We show the results for our CA model in Figure 5 and for the FIX model in Figure 8. Strikingly, we find that very few examples are actually required to train a working saliency map model. In particular, training on just 10 examples per class produces a model that gets only 0.7 LE more than using all of the training data and only 2.7 more than the fully trained model.

On the other hand, we find that the diversity of examples across classes is a bigger contributor to performance than the number of examples per class. For instance, training on 10 examples across all 1,000 classes gets an LE of 38.5, which is lower than training on 125 examples across only 100 classes. A similar pattern can be observed in the PxAP results.

Above all, these few-shot results indicate that training an effective saliency map model can be significantly simpler and more economical than previously thought. Saliency methods that require training a separate model such as Dabkowski & Gal (2017) and Zolna et al. (2020) are cheap to run at inference, but require an expensive training procedure, compared to gradient-based saliency methods or methods involving a per-example optimization. However, if training a masking model can be a lightweight procedure as we have demonstrated, then using masking models to generate saliency maps can now be a cheap and effective model interpretability technique.

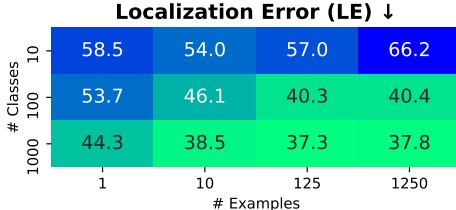 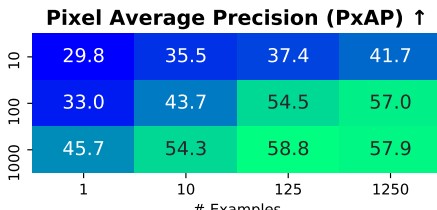

Figure 5: Training the saliency map models (CA) using fewer examples per class and/or fewer classes out of the 1000 in the training set. Results reported are the median over 5 runs with different samples of examples. Lower LE and higher PxAP are better. Models can be trained to generate good saliency maps with a surprisingly few number of training examples. For comparison, the LE and PxAP for the fully trained CA model are 35.8 and 59.4 respectively.

## 7 DISCUSSION AND CONCLUSIONS

In this work, we systematically evaluated the additive contribution of many proposed improvements to masking-based saliency map methods. Among the methods we tested, we identified that only the following factors meaningfully contributed to improved saliency map generation: (1) using both masked-in and masked-out objectives, (2) using the later layers of the classifier as input and (3) continually training the classifier. This simple setup outperforms other methods on WSOL metrics and PxAP, and passes a suite of saliency map sanity checks.

Strikingly, we also found that very few examples are actually required to train a saliency map model, and training with just 10 examples per class can achieve close to our best performing model. In addition, our masker model architecture is extremely simple: a two-layer ConvNet. Together, this suggests that learning masking-based saliency map extraction may be simpler than expected when given access to the internal representations of a classifier. This unexpected observation should make us reconsider both the methods for extracting saliency maps and the metrics used to evaluate them.

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

# A  MODEL DETAILS

## A.1  MASK REGULARIZATION

Without regularization, the masking model may learn to simply mask in or mask out the entire image, depending on the masking objective. We consider two forms of regularization. The first is L1 regularization over the mask pixels $m$, which directly encourages the masks to be small in aggregate. The second is Total Variation (TV) over the mask, which encourages smoothness:

$$\text{TV}(m) = \sum_{i,j}(m_{i,j} - m_{i,j+1})^2 + \sum_{i,j}(m_{i,j} - m_{i+1,j})^2,$$

where $i, j$ are pixel indices. TV regularization was found to be crucial by Fong & Vedaldi (2017) and Dabkowski & Gal (2017) to avoid adversarial artifacts. Hence, we have:

$$R(m) = \lambda_M \|m\|_1 + \lambda_{\text{TV}}\text{TV}(m). \tag{1}$$

Following Zolna et al. (2020), we only apply L1 mask regularization if we are using masked-in objective and the masked-in image is correctly classified, or we have a masked-out objective and the masked-out image is incorrectly classified–otherwise, no L1 regularization is applied for that example. In cases where we have both masked-in and masked-out objective, we have separate $\lambda_{M,in}$ and $\lambda_{M,out}$ regularization coefficients.

## A.2  CONTINUAL TRAINING OF THE CLASSIFIER

We largely follow the setup for training classifier-agnostic (CA) models from Zolna et al. (2020). Notably, when training on the masker objectives, we update $\theta_M$ but note $\theta_F$, to prevent the classifier from being optimized on the masker's objective. We maintain a pool of 30 different classifier weights in our classifier pool. We point the reader to Zolna et al. (2020) for more details.

## A.3  HYPERPARAMETERS

We show in Table 2 the space of hyperparameter search and hyperparameters for the best results, as show in the Table 1 in the main paper. We performed a random search over $\lambda_{M,out}, \lambda_{M,in}$, and $\lambda_{\text{TV}}$.

Aside from the hyperparameters shown in Table 2, we used a learning rate of 0.001 and a batch size of 72. We trained for 3.4 epochs (17 epochs on 20% of data) for all Train-Validation experiments and for 12 epochs for the Validation set experiments. Likewise, we use a learning rate decay of 5 for Train-Val experiments and 20 for Validation set experiments. For dual objective models, we used $\lambda_{\text{out}} = \lambda_{\text{in}} - 0.5$. We use the Adam optimize with 0.9 momentum and 1e-4 weight decay.

| Model | | Hyperparameters |
|---|---|---|
| a) FIX | + MaxEnt (O) | $\lambda_{M,out} = \{1, \underline{5}, 10, 15, 30\}\ \lambda_{TV} = \{0, 0.0001, 0.001, 0.005, \underline{0.01}, 0.05\}$ |
| b) FIX | + MinClass (O) | $\lambda_{M,out} = \{1, 5, 10, \underline{15}, 30\}\ \lambda_{TV} = \{0, 0.0001, 0.001, 0.005, 0.01, \underline{0.05}\}$ |
| c) FIX | + MaxClass (I) | $\lambda_{M,out} = \{1, 5, 10, \underline{15}, 30\}\ \lambda_{TV} = \{0, 0.0001, 0.001, 0.005, 0.01, \underline{0.05}\}$ |
| d) CA | + MaxEnt (O) | $\lambda_{M,out} = \{1, 5, \underline{10}, 15, 30\}\ \lambda_{TV} = \{\underline{0}, 0.0001, 0.001, 0.005, 0.01, 0.05\}$ |
| e) CA | + MinClass (O) | $\lambda_{M,out} = \{1, 5, 10, \underline{15}, 30\}\ \lambda_{TV} = \{0, 0.0001, \underline{0.001}, 0.005, 0.01, 0.05\}$ |
| f) CA | + MaxClass (I) | $\lambda_{M,out} = \{\underline{1}, 5, 10, 15, 30\}\ \lambda_{TV} = \{\underline{0}, 0.0001, 0.001, 0.005, 0.01, 0.05\}$ |
| g) FIX | + MaxClass (I) + MaxEnt (O) | $\lambda_{M,out} = \{1, 5, \underline{10}, 15\},\ \lambda_{M,in} = \{\underline{1}, 5, 10, 15\},$ $\lambda_{TV} = \{0, 0.0001, \underline{0.001}, 0.005, 0.01, 0.05\}$ |
| h) FIX | + MaxClass (I) + MinClass (O) | $\lambda_{M,out} = \{1, 5, 10, \underline{15}\},\ \lambda_{M,in} = \{1, 5, 10, \underline{15}\},$ $\lambda_{TV} = \{0, 0.0001, 0.001, 0.005, 0.01, \underline{0.05}\}$ |
| i) CA | + MaxClass (I) + MaxEnt (O) | $\lambda_{M,out} = \{1, 5, 10, \underline{15}\},\ \lambda_{M,in} = \{\underline{1}, 5, 10, 15\},$ $\lambda_{TV} = \{0, 0.0001, \underline{0.001}, 0.005, 0.01, 0.05\}$ |
| j) CA | + MaxClass (I) + MinClass (O) | $\lambda_{M,out} = \{1, 5, 10, \underline{15}\},\ \lambda_{M,in} = \{\underline{1}, 5, 10, 15\},$ $\lambda_{TV} = \{0, 0.0001, \underline{0.001}, 0.005, 0.01, 0.05\}$ |
| k) FIX | + MaxClass (I) + MaxEnt (O) + Layer[1] | $\lambda_{M,out} = 10, \lambda_{M,in} = 1, \lambda_{TV} = 0.001$ |
| l) FIX | + MaxClass (I) + MaxEnt (O) + Layer[3] | $\lambda_{M,out} = 10, \lambda_{M,in} = 1, \lambda_{TV} = 0.001$ |
| m) FIX | + MaxClass (I) + MaxEnt (O) + Layer[5] | $\lambda_{M,out} = 10, \lambda_{M,in} = 1, \lambda_{TV} = 0.001$ |
| n) FIX | + MaxClass (I) + MaxEnt (O) + Layer[4–5] | $\lambda_{M,out} = 10, \lambda_{M,in} = 1, \lambda_{TV} = 0.001$ |
| o) CA | + MaxClass (I) + MaxEnt (O) + Layer[1] | $\lambda_{M,out} = 15, \lambda_{M,in} = 1, \lambda_{TV} = 0.001$ |
| p) CA | + MaxClass (I) + MaxEnt (O) + Layer[3] | $\lambda_{M,out} = 15, \lambda_{M,in} = 1, \lambda_{TV} = 0.001$ |
| q) CA | + MaxClass (I) + MaxEnt (O) + Layer[5] | $\lambda_{M,out} = 15, \lambda_{M,in} = 1, \lambda_{TV} = 0.001$ |
| r) CA | + MaxClass (I) + MaxEnt (O) + Layer[4–5] | $\lambda_{M,out} = 15, \lambda_{M,in} = 1, \lambda_{TV} = 0.001$ |
| s) FIX | + MaxClass (I) + MaxEnt (O) + Layer[4–5] + Inf[Blur] | $\lambda_{M,out} = 10, \lambda_{M,in} = 1, \lambda_{TV} = 0.001$ |
| t) FIX | + MaxClass (I) + MaxEnt (O) + Layer[4–5] + Inf[CAG] | $\lambda_{M,out} = 10, \lambda_{M,in} = 1, \lambda_{TV} = 0.001$ |
| u) FIX | + MaxClass (I) + MaxEnt (O) + Layer[4–5] + Inf[DFN] | $\lambda_{M,out} = 10, \lambda_{M,in} = 1, \lambda_{TV} = 0.001$ |
| v) CA | + MaxClass (I) + MaxEnt (O) + Layer[4–5] + Inf[Blur] | $\lambda_{M,out} = 15, \lambda_{M,in} = 1, \lambda_{TV} = 0.001$ |
| w) CA | + MaxClass (I) + MaxEnt (O) + Layer[4–5] + Inf[CAG] | $\lambda_{M,out} = 15, \lambda_{M,in} = 1, \lambda_{TV} = 0.001$ |
| x) CA | + MaxClass (I) + MaxEnt (O) + Layer[4–5] + Inf[DFN] | $\lambda_{M,out} = 15, \lambda_{M,in} = 1, \lambda_{TV} = 0.001$ |
| y) FIX | + MaxClass (I) + MaxEnt (O) + Layer[4–5] + Inf[CAG] + GS | $\lambda_{M,out} = 10, \lambda_{M,in} = 1, \lambda_{TV} = 0.001, \tau = \{0.01, 0.05, 0.1, \underline{0.5}\}, GS = \{s, \underline{h}\}$ |
| z) CA | + MaxClass (I) + MaxEnt (O) + Layer[4–5] + Inf[CAG] + GS | $\lambda_{M,out} = 15, \lambda_{M,in} = 1, \lambda_{TV} = 0.001, \tau = \{0.01, 0.05, 0.1, \underline{0.5}\}, GS = \{\underline{s}, h\}$ |
| | *Validation Set* | |
| G) Our Best FIX | | $\lambda_{M,out} = 10, \lambda_{M,in} = 1, \lambda_{TV} = 0.001$ |
| H) Our Best CA | | $\lambda_{M,out} = 15, \lambda_{M,in} = 1, \lambda_{TV} = 0.001$ |

Table 2: Hyperparameter search space and chosen hyperparameters. Sets of values in a row indicate a grid search over all combinations in that row. Where there is a search, the hyperparameters for the best model, corresponding to results shown in Table 1, are underlined.

# B  SUPPLEMENTARY RESULTS

## B.1

## B.2  VARYING OBSERVED CLASSIFIER LAYERS

We show the full set of per-layer and layer-combination results in Table 4.

## B.3  COUNTERFACTUAL INFILLING AND BINARY MASKS

We show examples of saliency maps generated with counterfactual infillers in Figure 6 and incorporation of Gumbel-Softmax to generate masks with binary pixel values in Figure 7.

## B.4  SANITY CHECKS

We show in Table 5 the results for the Data Randomization Test (DRT) proposed by Adebayo et al. (2018). Here, we train a new classifier on the same training data but with labels shuffled across all images. We find that the similarity of saliency maps generated given a regularly trained classifier compared to those given a classifier trained on shuffled labels is low, indicating that the saliency maps reflect information learned from a well-formed image classification task.

## B.5  FEW-SHOT EXPERIMENTS

We show in Figure 8 the results for few-shot experiments using the FIX model configuration. We similarly find that very few examples are needed to train a good saliency map model.

| Model | | | | | F1 ↑ | Avg Mask |
|---|---|---|---|---|---|---|
| *Train-Validation Set* | | | | | | |
| a) FIX | + MaxEnt (O) | | | | 60.6± 0.1 | 56.3± 1.7 |
| b) FIX | + MinClass (O) | | | | 34.2± 10.2 | 31.1± 16.5 |
| c) FIX | + MaxClass (I) | | | | 21.6± 0.1 | 11.2± 0.1 |
| d) CA | + MaxEnt (O) | | | | 63.1± 0.0 | 51.1± 0.2 |
| e) CA | + MinClass (O) | | | | 64.1± 0.1 | 50.7± 0.3 |
| f) CA | + MaxClass (I) | | | | 49.2± 11.9 | 48.6± 11.9 |
| g) FIX | + MaxClass (I) + MaxEnt (O) | | | | 53.6± 0.1 | 35.2± 0.1 |
| h) FIX | + MaxClass (I) + MinClass (O) | | | | 30.7± 7.2 | 27.5± 16.0 |
| i) CA | + MaxClass (I) + MaxEnt (O) | | | | 62.9± 0.1 | 46.6± 0.3 |
| j) CA | + MaxClass (I) + MinClass (O) | | | | 64.8± 0.1 | 52.9± 0.3 |
| k) FIX | + MaxClass (I) + MaxEnt (O) | + Layer[1] | | | 57.7± 0.1 | 85.5± 1.3 |
| l) FIX | + MaxClass (I) + MaxEnt (O) | + Layer[3] | | | 53.4± 0.0 | 51.3± 0.1 |
| m) FIX | + MaxClass (I) + MaxEnt (O) | + Layer[5] | | | 57.9± 0.0 | 40.9± 0.1 |
| n) FIX | + MaxClass (I) + MaxEnt (O) | + Layer[4–5] | | | 56.1± 0.1 | 38.0± 0.1 |
| o) CA | + MaxClass (I) + MaxEnt (O) | + Layer[1] | | | 1.3± 0.0 | 0.8± 0.0 |
| p) CA | + MaxClass (I) + MaxEnt (O) | + Layer[3] | | | 55.9± 0.9 | 54.9± 2.0 |
| q) CA | + MaxClass (I) + MaxEnt (O) | + Layer[5] | | | 63.1± 0.1 | 48.6± 0.3 |
| r) CA | + MaxClass (I) + MaxEnt (O) | + Layer[4–5] | | | 63.0± 0.1 | 46.3± 0.0 |
| s) FIX | + MaxClass (I) + MaxEnt (O) | + Layer[4–5] | + Inf[Blur] | | 54.5± 0.1 | 36.6± 0.1 |
| t) FIX | + MaxClass (I) + MaxEnt (O) | + Layer[4–5] | + Inf[CAG] | | 55.7± 0.1 | 42.4± 0.1 |
| u) FIX | + MaxClass (I) + MaxEnt (O) | + Layer[4–5] | + Inf[DFN] | | 52.1± 0.0 | 32.5± 0.1 |
| v) CA | + MaxClass (I) + MaxEnt (O) | + Layer[4–5] | + Inf[Blur] | | 60.7± 0.1 | 42.1± 0.2 |
| w) CA | + MaxClass (I) + MaxEnt (O) | + Layer[4–5] | + Inf[CAG] | | 61.9± 0.1 | 46.7± 0.1 |
| x) CA | + MaxClass (I) + MaxEnt (O) | + Layer[4–5] | + Inf[DFN] | | 53.2± 0.2 | 32.7± 0.2 |
| y) FIX | + MaxClass (I) + MaxEnt (O) | + Layer[4–5] | + Inf[CAG] | + GS | 59.7± 0.2 | 57.2± 0.3 |
| z) CA | + MaxClass (I) + MaxEnt (O) | + Layer[4–5] | + Inf[CAG] | + GS | 57.7± 0.1 | 39.1± 0.2 |

Table 3:  Evaluation of masking-based saliency map methods. Each block captures one set of experiments. FIX indicates a fixed classifier, CA (Classifier-Agnostic) indicates training against a pool of continually trained classifiers. MaxClass (I), MinClass (O) and MaxEnt (O) are masked-in classification-maximization, masked-out classification-minimization and masked-out entropy maximization objectives for the masker. Layer[·] indicates the layer or layers of classifier activations provided as input to the masker.  Inf[·] indicates the infiller operation applied after masking–the default otherwise is no infilling. Columns show mean and standard errors over 5 runs for F1 and the average mask magnitude. Underlined results are the best results within that block, while **bold** are the best results for data set, excluding baselines.

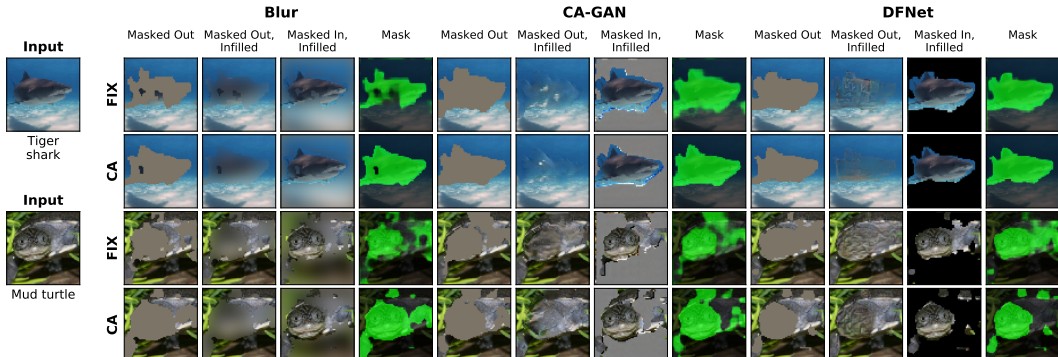

Figure 6:  Saliency maps using various infilling methods for counterfactual generation. Following Chang et al. (2019); Agarwal & Nguyen (2019), we infill masked out portions of the image and provide the resulting infilled image to the classifier. Infillers are hypothesized to help training by making the classifier inputs look closer to natural images as well as forcing the masker to mask out all evidence of a salient object. FIX indicates a fixed classifier, CA (Classifier-Agnostic) indicates training against a pool of continually trained classifiers. We do not find quantitative improvements from incorporating an infiller.

| Model | | | ‖ OM ↓ | LE ↓ | F1 ↑ | SM ↓ | PxAP ↑ | ‖ Mask |
|---|---|---|---|---|---|---|---|---|
| **Train-Validation Set** | | | | | | | | |
| a) FIX | + MaxClass (I) + MaxEnt (O) | + Layer[1] | 57.3 | 48.2 | 57.6 | 0.40 | 34.94 | 84.0 |
| b) FIX | + MaxClass (I) + MaxEnt (O) | + Layer[2] | 55.8 | 46.4 | 53.5 | 0.37 | 38.26 | 61.5 |
| c) FIX | + MaxClass (I) + MaxEnt (O) | + Layer[3] | 53.6 | 43.9 | 53.5 | 0.31 | 43.97 | 51.6 |
| d) FIX | + MaxClass (I) + MaxEnt (O) | + Layer[4] | 47.9 | 36.9 | 55.2 | 0.15 | 58.83 | 41.2 |
| e) FIX | + MaxClass (I) + MaxEnt (O) | + Layer[5] | 48.5 | 37.9 | 57.9 | 0.04 | 55.53 | 40.9 |
| f) FIX | + MaxClass (I) + MaxEnt (O) | + Layer[2–5] | 46.7 | 35.8 | 53.6 | 0.08 | 56.75 | 35.5 |
| g) FIX | + MaxClass (I) + MaxEnt (O) | + Layer[3–5] | 46.7 | 35.7 | 54.6 | 0.06 | 57.53 | 36.2 |
| h) FIX | + MaxClass (I) + MaxEnt (O) | + Layer[4–5] | 47.0 | 36.1 | 56.0 | 0.04 | 58.11 | 37.8 |
| i) CA | + MaxClass (I) + MaxEnt (O) | + Layer[1] | 77.5 | 71.9 | 1.3 | 0.49 | 27.24 | 0.8 |
| j) CA | + MaxClass (I) + MaxEnt (O) | + Layer[2] | 71.9 | 65.1 | 1.3 | 0.62 | 27.25 | 0.8 |
| k) CA | + MaxClass (I) + MaxEnt (O) | + Layer[3] | 54.9 | 45.1 | 56.2 | 0.27 | 44.51 | 54.9 |
| l) CA | + MaxClass (I) + MaxEnt (O) | + Layer[4] | 46.7 | 35.1 | 60.8 | 0.16 | 57.94 | 48.2 |
| m) CA | + MaxClass (I) + MaxEnt (O) | + Layer[5] | 46.4 | 35.2 | 63.2 | 0.11 | 56.95 | 48.5 |
| n) CA | + MaxClass (I) + MaxEnt (O) | + Layer[2–5] | 45.4 | 33.6 | 63.1 | 0.13 | 60.88 | 47.2 |
| o) CA | + MaxClass (I) + MaxEnt (O) | + Layer[3–5] | 45.3 | 33.4 | 63.3 | 0.12 | 61.47 | 47.1 |
| p) CA | + MaxClass (I) + MaxEnt (O) | + Layer[4–5] | 45.0 | 33.2 | 63.1 | 0.11 | 61.24 | 46.4 |

Table 4: Evaluation of masking-based saliency map methods, varying the layers provided to the masker. FIX indicates a fixed classifier, CA (Classifier-Agnostic) indicates training against a pool of continually trained classifiers. MaxClass (I), MinClass (O) and MaxEnt (O) are masked-in classification-maximization, masked-out classification-minimization and masked-out entropy maximization objectives for the masker. Layer[·] indicates the layer or layers of classifier activations provided as input to the masker. Inf[·] indicates the infiller operation applied after masking–the default otherwise is no infilling. Columns show evaluation metrics Official Metric (OM), Localization Error (LE) and F1 for weakly supervised localization, Saliency Metric (SM) and Pixel Average Precision (PxAP). Mask indicates numerical average of masking pixel values. Underlined results indicates the best results within that block, while **bold** indicates best results for data set, excluding baselines.

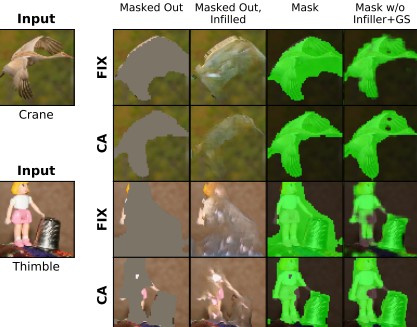

Figure 7: We apply the Gumbel-Softmax trick to train the masker to produce binary masks, infilling using the CA-GAN. As seen in Figure 2, most models in our experiments produce mask pixel values of 0 or 1, so the benefits of explicitly learning a discrete output distribution are limited.

| Model | Rank Correl(Abs) | Rank Correl(No Abs) | HOGS Similarity | SSIM |
|---|---|---|---|---|
| FIX | -0.069 | 0.037 | 0.488 | 0.001 |
| CA | 0.129 | 0.133 | 0.519 | 0.022 |

Table 5: Data Randomization Test. The saliency map methods are applied to both a regular classifier, as well as a classifier trained on randomly shuffled labels, and the similarity of the generated saliency maps are measured. Both FIX and CA methods show low similarity between the saliency maps generated by regular and shuffled classifiers.

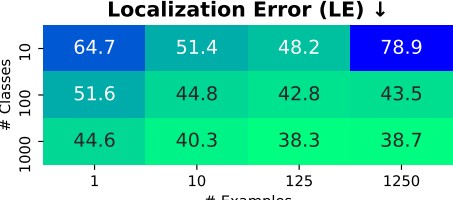 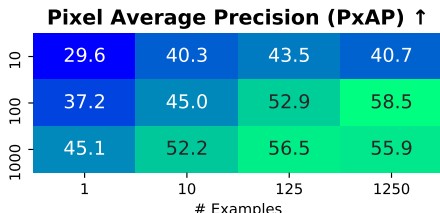

Figure 8: Training the saliency map models (FIX) using fewer examples per class and/or fewer classes out of the 1000 in the training set. Results reported are the median over 5 runs with different samples of examples. Lower LE and higher PxAP are better. Models can be trained to generate good saliency maps with a surprisingly few number of training examples.

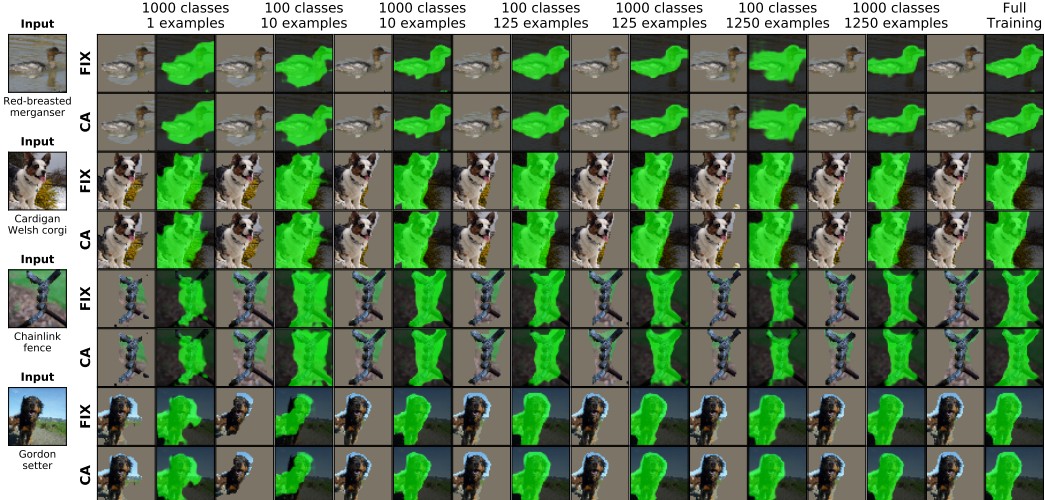

Figure 9: Examples of saliency maps computing from models trained with less data. Columns correspond to the models shown in Figure 5 and Figure 8. Even models trained with very few examples per class produce saliency maps similar to the fully trained model.

