# OpenReview forum: "Investigating and Simplifying Masking-based Saliency Methods for Model Interpretability"
_ICLR.cc/2021/Conference — Reject_

### Official Review · AnonReviewer4 · 2020-10-26
**The proposed method is simpler and better than the previous methods, but my concerns are the limited novelty and no significant improvement**

**Rating:** 6
**Confidence:** 4

**Review:**

/*************************Post-Rebuttal**********************/
The authors address many of my concerns well, and I agree with their rebuttal.

The modified manuscript also looks good, too.

I raise my rating.

/*************************Pre-Rebuttal**********************/

Pros.:
1. The proposed method is simple and reasonable but better than the current state-of-the-art methods.
2. The paper is clearly presented and well-organized.
4. The authors conduct many ablation studies to validate the different variants of the proposed methods.
5. The few-shot localization built in the proposed method is also interesting, and worth being explored.


Cons.:
1. Limited Novelty

The proposed method is similar to [Ref. 1, 2], but there is no discussion in the related work. After reading this paper and [Ref. 1, 2], I don't see any significant difference in [Ref. 1, 2], the authors focus on top-down saliency detection, and use the masked image scheme to train the network. They proposed a CNN framework that contains two CNN modules, an image-level classifier and a pixel-level map generator. They also use the saliency from the map generator as the classifier's input and then optimize the classifier scores to derive the saliency map and train the generator. In this paper, the loss is L2 loss and binary-class setting, but I think the main idea is almost the same. Therefore, I think the novelty is very limited. I would like to see that the authors point the significant difference between the proposed method and [Ref. 1, 2] and give a detailed discussion.

[Ref. 1] Hsu et al., Weakly Supervised Saliency Detection with A Category-Driven Map Generator, BMVC'17

[Ref. 2] Hsu et al., Weakly Supervised Salient Object Detection by Learning A Classifier-Driven Map Generator, TIP'19

2. No significant improvement

Compared to Zolna et al. (2020), the improvement of the proposed method is not significant, 48.6 v.s. 48.4 in OM,  36.1 v.s. 35.8 in LE, 61.4 v.s. 62.0 in F1. I think this improvement is minor.

3. About few-shot experiments

The few-shot experiments are interesting, but why aren't other methods conducted for these experiments? The authors claim that the few samples are sufficient to train the proposed method, but other methods may also require a small number of training samples to achieve similar performance. Besides, I am not clear why the proposed method could work well under the few-shot setting because there is no specific module or design for the few-shot setting.   Could the authors explain it in detail?

4. Other issue:

What is the difference between the proposed method and other weakly supervised or unsupervised object saliency detection, such as [Ref 3, 4, 5, 6, 7]? In these works, their goal is also to train a network to predict a class-agnostic saliency map and highlight the most salient object in the images under the unsupervised or weakly supervised setting. Therefore, I would like to know the major difference between the tasks. If they are similar to the solved problem, the discussion and comparison should be conducted.

[Ref. 3] Zhang et al., Zhang. Supervision by fusion: Towards unsupervised learning of deep salient object detector, ICCV'17

[Ref. 4] Li et al., Weakly supervised salient object detection using image labels, AAAI'19

[Ref. 5] Wang et al., Learning to detect salient objects with image-level supervision, CVPR'17

[Ref. 6] Zhang et al., Deep unsupervised saliency detection: A multiple noisy labeling perspective, CVPR'18

---

> ### Author Response · Authors · 2020-11-23
> **Response #4 (Part 2)**
>
>
> > 3. About few-shot experiments
> > The few-shot experiments are interesting, but why aren't other methods conducted for these experiments? The authors claim that the few samples are sufficient to train the proposed method, but other methods may also require a small number of training samples to achieve similar performance. Besides, I am not clear why the proposed method could work well under the few-shot setting because there is no specific module or design for the few-shot setting. Could the authors explain it in detail?
>
> We highlight that our presentation of the few-shot results does not argue that our method does best in a few-shot setting, but rather that: given a good saliency method (such as the one we propose), one only needs a small number of examples to get surprisingly good saliency maps. Indeed, the effectiveness of the few-shot experiments was somewhat surprising for us as well. We highlight that having a saliency method work with very few examples of training is itself not entirely unexpected - methods such as guided-backprop and Grad-CAM, while performing quite a bit worse, are effectively zero-shot saliency methods (requiring no training for saliency map generation). We see the few-shot experiments we have as a midpoint between fully-trained saliency methods such as Hsu et al., (2017) and Dabkowski and Gal, (2018), and the zero-shot methods such as those highlighted above that more directly exploit the trained image classifier. We agree that few-shot experiments with the other methods compared to would also be informative.
>
> > 4. Other issue:
> > What is the difference between the proposed method and other weakly supervised or unsupervised object saliency detection, such as [Ref 3, 4, 5, 6, 7]? [...]
>
> As discussed above, we see research on saliency map generation falling largely into two camps: saliency object detection as a goal in itself (identifying salient objects in a given image), and saliency maps as a means of model interpretability (identifying regions deemed salient by a given image classification-model). The four works mentioned fall into the former category, while we conceived of our work from the perspective of the latter. We agree that both lines of research have a great deal of common ground: many of these methods are now using pretrained image-classification models such as ResNets as their backbone. Similarly, our CA model continues to train the classifier alongside the masker, hence ostensibly focusing less on interpreting a specific given model. (a slightly subtle point: the masker can be trained to be classifier-agnostic, but applied to the original ResNet-50 classifier at evaluation). At the same time, there are also differences between the goals of both lines of work: as discussed above, techniques that improve saliency maps that are orthogonal to the classifier would fall out of scope for model interpretability. Given your comment, we agree that we should have recontextualized our work with regards to both salient object detection and saliency-as-interpretability fields rather than just the latter, and have adjusted our related work and discussion accordingly.

---

> ### Author Response · Authors · 2020-11-23
> **Response #4 (Part 1)**
>
> Thank you for your response. To address a broad theme in the response, we highlight that the method we are proposing is not intended to be novel - rather, we are empirically evaluating, ablating and combining existing ideas in the literature to distill which proposed contributions materially improve performance.
>
> Addressing your points:
>
> > 1. Limited Novelty
> > The proposed method is similar to [Ref. 1, 2], but there is no discussion in the related work. [...]
>
> Thank you for highlighting the cited works - we have unfortunately missed these works in our literature review, and we have updated our manuscript to cite them. To elaborate briefly on the comparison and distinctions here:
> - As mentioned above, the goal of the work is to empirically compare and ablate existing ideas in the literature, rather than propose new objectives or methods to improve saliency map generation.
> - Secondly, we see a distinction between salient object detection as an end in itself, and saliency-map generation for model interpretability. [1, 2] fall within the first category, whereas our work falls within the latter. While there are similarities between both lines of work, they have different focuses and different problem constraints. For instance, work in the former category may apply techniques to refine saliency maps that are unrelated to the image classifier (e.g. extracting superpixels and using external object proposals in [2]), whereas these would fall out of scope for methods for model interpretability. Overall, it is difficult to directly compare our work and those of [1,2] in terms of results because of the differences in training regime and evaluation tasks, although we acknowledge they are definitely within the same scope of ideas for training and regularizing models for generating saliency maps. An extended version of this work could feasibly evaluate the ideas proposed therein as well (e.g. the relative contribution of training on a separate D_bg, vis-à-vis the training objectives we already have, using superpixels).
>
>
> > 2. No significant improvement
> > Compared to Zolna et al. (2020), the improvement of the proposed method is not significant, 48.6 v.s. 48.4 in OM, 36.1 v.s. 35.8 in LE, 61.4 v.s. 62.0 in F1. I think this improvement is minor.
>
> We agree that our bottom-line results on WSOL are only a marginal improvement over Zolna et al. (2020). We would like to highlight that the primary contribution of our work is the extensive set of experiments evaluating different methods and training objectives in Table 1, and identifying which proposed improvements materially contribute to improved performance. The evaluation on the test set, and the outperformance of Zolna et al. primarily serves to validate our findings. In fact, the observation that multiple additional elements of the model bring relatively small benefits is what we believe to be one of the most important findings of this paper.

---

### Official Review · AnonReviewer2 · 2020-10-28
**Thorough experimentation and important insight on the matter of deriving saliency from trained classifiers**

**Rating:** 7
**Confidence:** 3

**Review:**

Summary:

By the first look, this work itself does not introduce any new architecture or novel algorithm. It takes what is considered as the popular choices in generating classifier saliency masks, and conducts quite extensive sets of experiments to dissect the components by their importance. The writing is pretty clear in narrative and the experimental findings are surprising and significant.

Good things to mention:

1. The importance of using CA seems quite evident from Table 1.
2. The importance of using multiple resolutions is also evident.
3. The findings of few shot learning ability for masker is quite interesting, class diversity seems to play a very import role.

Some questions:
1. Given the fact that all experiments are done with ImageNet, which is widely considered as a solved task with almost super-human classifiers, do you think the superb performance of those classifiers could inflate your findings? Do you think the results in Table 1 and Figure 5 would still hold consistently when the classifier is not as performant?
2. I am not sure I understand the point of the figure on the right of Figure 4. What is the take-way message from it?
3. Compare (e,f,i,j) in Table 1, it seems I does not make a difference as long as O is used. What is the reason there?

---

> ### Author Response · Authors · 2020-11-23
> **Response #2**
>
> Thank you for your response. Addressing your questions:
>
> > 1. Given the fact that all experiments are done with ImageNet, which is widely considered as a solved task with almost super-human classifiers, do you think the superb performance of those classifiers could inflate your findings? Do you think the results in Table 1 and Figure 5 would still hold consistently when the classifier is not as performant?
>
> It certainly is the case that the methods and performance metrics are heavily dependent on the dataset and models that we are evaluating on. How the performance of the underlying classifier interacts with the performance of these methods is a very important question, and one that we would absolutely be interested in further investigating (particularly as such saliency methods are applied to other domains, such as medical imaging). While it is difficult to extrapolate from the current results, we hypothesize that the results would remain qualitatively similar - the later stages of ImageNet training may be refining the classification of objects within broad categories (e.g. different kinds of fish), whereas the primary challenge of saliency map generation is to simply identify the most salient region. However, the story might be very different for a classifier that is significantly struggling with a dataset, e.g. relying on background textures rather than the salient/foreground object.
>
> > 2. I am not sure I understand the point of the figure on the right of Figure 4. What is the take-way message from it?
>
> Figure 4 shows the results of the Model Parameter Randomization Test, a sanity check for saliency maps proposed by Adebayo et al. We were not able to elaborate/introduce it in as much depth as we would like in the prose due to length constraints, but the figure is close to those shown in Adebayo et al. As such, we have extended the discussion for the MPRT in the paper and DRT in the appendix. The broad take-away is that the saliency maps get significantly distorted as we cumulatively randomize the parameters of the underlying classifier, passing the sanity check they propose.
>
> > 3. Compare (e,f,i,j) in Table 1, it seems I does not make a difference as long as O is used. What is the reason there?
>
> We are a little unclear about this question, so let us know if we are not addressing it directly and we will be happy to follow up. Having a masked-in (I) objective does contribute to improved performance even when there is a masked-out (O). For instance: row (j) outperforms row (e). On the other hand, masked-out objectives appear to help much more than masked-in objectives; a potential intuitive explanation for this is that a masked-in objective encourages the model to only select the minimal features for the classifier to successfully classify an image (e.g. dog’s ears), whereas the masked-out objective encourages the masked to mask out all regions that could help the classifier (e.g. any parts of a dog’s body help the classifier, so the masker will try to ask the whole dog out). Ultimately, we find that combining both masked-out entropy-maximization and a masked-in classification-minimization objects leads to the best performance.

---

### Official Review · AnonReviewer1 · 2020-10-29
**Unsupported claims**

**Rating:** 4
**Confidence:** 5

**Review:**

The paper proposes an approach to improve masked based prediction explanation. They train an auxiliary model which predicts a mask that must satisfy two terms 1) maximize classification accuracy when applied to the image and 2) maximize entropy over softmax when the inverse mask is applied to the image (they also experiment with minimizing the classification accuracy instead here).

The paper also positions their work to be classifier agnostic in the text which is not clear to me. I think this aspect is a negative because it is not in the list of contributions and seems like a distraction of a remnant of an old direction of the paper:

> "Whereas a FIX approach seeks a saliency map that explains what regions are most salient to a given classifier, a CA approach tries to identify all possible salient regions for any hypothetical classifier (hence, classifier-agnostic). In other words, a CA approach may be inadequate for interpreting a specific classifier and is better suited for identifying salient regions for a class of image classification models."

I'll focus now on studying how this paper addresses the 4 contributions they claim in the intro:

> (1) We find that incorporating both masked-in classification maximization and masked-out entropy maximization objectives leads to the best saliency maps, and continually training the classifier improves the quality of generated maps.

There are no standard deviations reported from multiple runs so nothing can be statistically claimed. Also, looking at table 1 this is not clear. It seems MinClass works the best compared to almost all other methods in terms of PxAP. In terms of other metrics it seems like no best can be determined.

> (2) We find that the masking model requires only the top layers of the classifier to effectively generate saliency maps.

This is reported in the table. It is not clear if this is a strong contribution as it would just be specific to this method and without a standard deviation we cannot conclude anything.

> (3) Our final model outperforms other masking-based methods on WSOL and PxAP metrics.

The reported difference between the methods is 48.6 vs 48.4.

> (4) We find that a small number of examples—as few as ten per class—is sufficient to train a masker to within the ballpark of our best performing model.
This is reported but the paper doesn't have a section detailing the experiments or showing how this number is derived.


The paper could be improved by refining the contributions and detailing what evidence should be observed to support these claims and then providing a significant amount of evidence. Right now the paper is not focused in general and does not focus on supporting the claims made in the introduction and therefore is not ready for publication.

---

> ### Author Response · Authors · 2020-11-23
> **Response #1**
>
> Thank you for your response.
>
> Addressing your first query, the “classifier-agnostic” model is largely derived from Zolna et al., (2020). In their formulation (which we follow), the goal is to train a masker to generate masks that will not just confuse a single classifier (wherein the masker may simply learn adversarial perturbations or artifacts to do so), but any classifier within a class of models. They approximate this by continually training the classifier and sampling from its past history of weights to simulate a diverse pool of classifiers, and train the masker to fool all of them, alternately between training the masker and classifier. As such, the masker is deemed to be “classifier-agnostic” i.e. not targeting a single given classifier. We follow their naming convention.
>
> Addressing your subsequent points:
>
> > There are no standard deviations reported from multiple runs so nothing can be statistically claimed. Also, looking at table 1 this is not clear. It seems MinClass works the best compared to almost all other methods in terms of PxAP. In terms of other metrics it seems like no best can be determined.
>
> We agree that standard deviations across multiple runs would significantly strengthen the intellectual contribution of our results. As such, we reran the experiments shown in Table 1 an additional 4 times, and present the standard deviation computed over the 5 samples in the updated Table 1. To directly address the reviewer’s concerns: the standard errors for the models that perform well are relatively small, and should be sufficient to distinguish relative performance of the different model configurations.
>
> > This is reported in the table. It is not clear if this is a strong contribution as it would just be specific to this method and without a standard deviation we cannot conclude anything.
>
> While we agree that performing the same layer-wise variation experiment across more model configurations would clarify the finding, we highlight that this observation is consistent across both the FIX and CA models. We address the point on standard deviations above.
>
> > The reported difference between the methods is 48.6 vs 48.4.
>
> We highlight that the primary contribution of our work is the extensive set of experiments evaluating different methods and training objectives in Table 1, and identifying which proposed improvements materially contribute to improved performance. The evaluation on the test set, and the outperformance compared to Zolna et al. primarily serves to validate our findings. Moreover, a primary takeaway of our empirical study is that many of these proposed improvements in fact do not improve upon the basic model. While the margin of outperformance over the compared model is small, we further highlight that it is not uncommon for margins of improvement to shrink as the field matures, and WSOL is a task that has been extensively hill-climbed on.
>
> > The paper could be improved by refining the contributions and detailing what evidence should be observed to support these claims and then providing a significant amount of evidence. Right now the paper is not focused in general and does not focus on supporting the claims made in the introduction and therefore is not ready for publication.
>
> We see our work as focusing on the following: evaluating the marginal contribution of recently proposed improvements to saliency-map-for-interpretability methods in a systematic manner, with a follow-up set of few-shot experiments based on our findings. To that end, we present a fairly extensive set of results (Table 1), each row of which includes its own hyperparameter search (Table 2). We have refined some of the wording in the paper to reflect the concerns raised in the comment.

---

> > ### Comment · AnonReviewer1 · 2020-11-24
> > **.**
> >
> > Thank you for addressing my points about running multiple iterations of the experiments.
> >
> > On the contribution of "evaluating the marginal contribution of recently proposed improvements to saliency-map-for-interpretability methods in a systematic manner" Why is this not the title then? I still feel the paper is not focused.
> >
> > However, I will raise my score based on your added experiments.

---

> > > ### Author Response · Authors · 2020-11-25
> > > **Response #1a**
> > >
> > > Thank you. We thought that the message of the paper was sufficiently clear and the title emphasizes the investigation of methods, but if you believe otherwise, we will work hard to make this clearer in the final version if the paper gets accepted.
> > >
> > > As mentioned above, we have conducted additional experiments to add standard errors to our results following your feedback. Besides adjusting how we framed the paper (which we are willing to work on), what do you think this paper is lacking that could put it over the threshold of acceptance?

---

### Official Review · AnonReviewer3 · 2020-10-29
**The paper has sufficient experiments on strategies of masking-based saliency methods , but lacks theoretical analysis.**

**Rating:** 6
**Confidence:** 2

**Review:**

-Summary:
This paper investigates the previous masking-based saliency map methods. By detailly formulate the masking-based saliency methods and sufficient experimentation, the paper gives a simple formulation and practical training strategies.

-Strength:
The paper is well organized and the presentation is easy to follow.
The experiments are comprehensive.

-Weakness:
Although the experiments are enough, the paper simply adopts the previous masking-based methods, evaluation metrics, architectures, and sanity check analysis. The theoretical analysis is insufficient.  More theoretical experiments instead of performance analysis need to be added.

---

> ### Author Response · Authors · 2020-11-23
> **Response #3**
>
> Thank you for your response. We absolutely agree that more theoretical analysis of why and how these saliency methods work would be incredibly valuable. On our part, we see our submission as an extensive empirical study comparing a broad range of proposed methods (each with their own theoretical or intuitive motivations) as well as metrics in a fairly compared setting. Jointly analyzing such a diverse set methods theoretically is remarkably challenging and has not been attempted to date to our knowledge. A first step towards such analysis would be to integrate different proposed approaches into a single framework as we have done. We hope that our results and findings can help to narrow the field’s focus on methods that materially improve performance. This way, by simplifying the existing approaches, we form a better basis for future theoretical investigation into masking-based saliency/interpretability methods.

---

### Author Response · Authors · 2020-11-23
**Overall Response**

We thank all four reviewers for the feedback and comments. They have been valuable for us to refine the text and content of our submission. We have made two major updates to our paper based on the feedback provided:

1. We have significantly expanded upon the results in Table 1 by computing the mean and standard errors for each model configuration. Each row in the Train-Validation section of Table 1 now reflects means and standard errors of metrics computed over 5 runs. For space reasons, we have moved the “F1” and “Mask” columns of the table to the appendix.

2. We have expanded our related work section to discuss work on salient object detection methods (methods that are not focused on model interpretability, but purely on saliency map generation).

We would further like to highlight that the main aim of the paper is to evaluate and identify which proposed improvements to a basic saliency map framework materially contribute to better performance, which we do so through an extensive empirical study. The novelty of our work lies in demonstrating that a surprisingly simple saliency map extraction method, using only a subset of popular ingredients is sufficient to achieve very strong performance, and this extends even to a few-shot setting. Improving upon the results in the literature is a just side effect of our investigation, which validates our experiment design.

We hope that these changes are able to address some of the points raised by the reviewers, and we are happy to continue the conversation.

---

### Decision · Program_Chairs · 2021-01-07
**Final Decision**

**Decision:**

Reject

**Comment:**

While the reviewers found parts of the paper interesting, the main concern about this paper was lack of novelty and marginal improvements obtained by the proposed methods.